# Combined Repair and Reconstruction of Coracoclavicular and Acromioclavicular Ligaments for Acute and Chronic AC Joint Dislocations: A Technical Note and Prospective Case Series

**DOI:** 10.3390/jcm14051730

**Published:** 2025-03-04

**Authors:** Freek Hollman, Mohammad Nedal Jomaa, Nagmani Singh, Roberto Pareyón, Helen M. A. Ingoe, Sarah L. Whitehouse, Rohit Mahesh Sane, Tristan Shuker, Kenneth Cutbush

**Affiliations:** 1Queensland Unit for Advanced Shoulder Research QUASR, Queensland University of Technology, Brisbane, QLD 4000, Australia; freekhollman@gmail.com (F.H.); dr.mohammadnedaljomaa@gmail.com (M.N.J.); s.whitehouse@qut.edu.au (S.L.W.);; 2NOA-Southside Clinical Unit, School of Medicine, Faculty of Health, Medicine and Behavioural Sciences, University of Queensland, Brisbane, QLD 4072, Australia

**Keywords:** AC joint reconstruction, AC joint repair, AC and CC ligament complex, acute and chronic AC joint dislocation, technical note

## Abstract

**Background/Objectives:** Dislocation of the acromioclavicular joint (ACJ) is a common injury for which numerous operative fixation and reconstructive techniques have been described. This technique combines a coracoclavicular ligament (CC) repair with an acromioclavicular ligament (AC) and CC reconstruction with an additional ACJ internal brace to address both horizontal and vertical instability. **Methods:** The surgery is performed through a superior approach in the following sequence: (1) CC ligaments are repaired using a TightRope construct, (2) CC reconstruction is performed using a peroneus longus tendon allograft, (3) AC ligaments are repaired using an internal brace, and (4) AC reconstruction is performed with a second peroneus longus tendon allograft. The results of consecutive patients with grade IIIB, IV, and V AC joint dislocations were included. **Results:** Six patients with acute and six patients with chronic injuries were eligible for inclusion. The Constant–Murley Score improved significantly from 27.6 (8.0–56.5) to 61.5 (42.0–92.0) (*p* = 0.006 paired *t*-test) at 12 months of follow-up. There was one complication (frozen shoulder) from which the patient recovered spontaneously; no other complications were observed with this technique. The coracoclavicular distance (CCD) was reduced from 18.7 mm (13.0–24.0) to 10.0 mm (6.0–16.0, *p* < 0.001) and 10.5 mm (8.0–14.0, *p* = 0.002) at 12 weeks and 12 months, respectively. **Conclusions:** This study describes a new technique to treat acute and chronic Rockwood stage IIIB–V ACJ dislocations with promising short-term clinical and radiological results. The results suggest that the combined repair and reconstruction of the AC and CC ligaments is a safe procedure with low complication risk in experienced hands. Addressing the vertical and horizontal stability in ACJ dislocation is key to achieving optimal long-term results. Further, follow-up is required to investigate the long-term outcomes.

## 1. Introduction

Injury to the acromioclavicular joint (ACJ) is common within the shoulder girdle, accounting for 9% of all shoulder injuries [1,2]. The incidence in the general population is described to be 3–4 cases per 100,000 persons per year [2]. The literature supports performing ACJ reconstruction or repair in Rockwood type IIIB, IV, V, and VI injuries [3,4]. Over the years, many treatment options have been described to reduce and stabilize the dislocated ACJ. Various definitions for acute and chronic tears have also been proposed over time, with a general consensus that there may be greater potential for tissue healing when the repair is performed acutely [5,6]. Understanding ACJ anatomy is paramount in order to properly address the clinical problem. The acromioclavicular (AC) capsule comprises superior, inferior, anterior, and posterior components. The superior and posterior ligaments are the strongest, contributing significantly to horizontal stability. The coracoclavicular ligament (CC) complex consists of the conoid and trapezoid ligaments that insert on the posteromedial and anterolateral region of the undersurface of the distal clavicle, respectively. The CC ligaments mainly serve to provide vertical stability, while the coracoacromial ligament is a strong triangular band that connects the coracoid process to the acromion and provides vertical stability as well [7,8,9].

Over 60 different surgical techniques have been published, indicating that there is no consensus or gold-standard treatment [10]. There is some evidence to suggest that techniques addressing both the CC ligaments and the capsule of the ACJ may more effectively restore ACJ stability compared to techniques that address only the CC ligament complex [11,12,13]. The majority of techniques described do not specifically repair or restore horizontal stability [10]. With the growing appreciation of the biomechanical significance of the AC ligaments for horizontal stability, techniques have been developed that combine CC repair with concomitant AC ligament repair [1,14,15,16,17]. These techniques have shown increased joint stability in the horizontal plane compared to isolated CC ligament repair [1,14,15,16,17]. When only the vertical component is addressed, a proportion of patients will continue to have posterior subluxation of the ACJ, with potential residual shoulder pain [12,18]. Additionally, isolated CC ligament repairs are often subject to high areas of strain around the construct, leading to hardware failure and fractures [19]. The technique described utilizes both drill holes and anchors and theoretically confers a similar risk of fractures compared to other techniques that weaken the bone due to multiple drill holes.

In this study, a surgical technique is described that can be used for both acute and chronic ACJ dislocations. The technique described in this paper includes an open repair combined with reconstruction of both the AC and CC ligaments. It is hypothesized that combining a CC and AC repair with a CC and AC reconstruction using a tendon allograft will result in high maintenance of anatomical reduction and improved functional results. This study provides a detailed description of the technique and a prospective case series with clinical and radiographic results.

## 2. Materials and Methods

### 2.1. Surgical Technique

#### 2.1.1. Approach

The patient is positioned in the beach chair position with the arm attached to an upper limb positioner (Spider, Smith & Nephew, Watford, UK). A routine diagnostic arthroscopy of the glenohumeral joint is performed using posterior and anterior portals to check for concomitant pathology. A superior approach to the clavicle and ACJ is performed through an incision placed parallel and anterior to the clavicle. The incision runs from the junction of the middle and lateral third of the clavicle to the lateral border of the tip of the acromion. All dissection is performed with electrocautery. The anterior deltoid is detached from the lateral third of the clavicle and anterior acromion (Figure 1A). The coracoid process is exposed. The pec minor insertion is released from the medial border of the coracoid, and the coracoacromial ligament is released from the coracoid’s lateral border. This allows for the smooth passing of the tendon allograft around the coracoid. The ACJ is debrided with bone nibblers, and the dislocation is reduced and fixed temporarily with a 2 mm K-wire (2.0 mm K-wire, Art. No 292.652, DePuy Synthes Companies, Raynham, MA, USA) passed from the lateral aspect of the acromion through the distal clavicle (Figure 1B). The distal clavicle was not routinely excised as in other techniques.

#### 2.1.2. CC Dog Bone Repair

With the ACJ reduced and pinned in the correct position, a cannulated drill (Cannulated Drill for AC Repair, 2.4 mm, AR-2257D-24, Arthrex, Naples, FL, USA) is passed from the superior surface of the clavicle through the clavicle and the coracoid so that it exits just below the inferior surface of the base of the coracoid process. This drill placement leads to a theoretically non-anatomical repair, being somewhere in between the conoid and trapezoid ligaments. The drill is cannulated, and once correctly in position, the obturator is removed, and a passing wire (Nitinol Suture Passing Wire, AR-1255-18, Arthrex, Naples, FL, USA) is placed down the cannulated drill from superior to inferior. The passing wire is retrieved from the tip of the drill inferior to the base of the coracoid with an artery forceps. Using the passing wire, two self-locking TightRope ABS (Attachable Button System) implants (TightRope ABS Implant with UHMWPE, AR-1588TN, Arthrex, Naples, FL, USA) are passed from inferior to superior through the drill hole in the coracoid and clavicle. Prior to passing the TightRopes under direct view, a Dog Bone implant (Dog Bone Button, AR-2270, Arthrex, Naples, FL, USA) is attached to the distal end of the TightRopes. As the ABS TightRopes are pulled into position, the Dog Bone button comes to lie against the inferior surface of the coracoid process, securing the inferior end of the TightRopes to the coracoid process (Figure 1B).

Once the superior ends of the ABS TightRopes have been retrieved from the drill hole in the superior surface of the clavicle, a second Dog Bone button is attached to the superior ends of the TightRopes. The TightRopes are sequentially tightened using the self-locking mechanism that is integral to the TightRopes to secure the second Dog Bone button on the superior surface of the clavicle at the correct tension. As the ACJ is held in a reduced position by the K-wire, the correct tension of the TightRopes can be achieved, and overtightening can be avoided.

#### 2.1.3. CC Ligament Reconstruction

After the Dog Bone and TightRope construct is secured, a tendon allograft, peroneus longus 310 mm (Peroneus Longus Tendon, AB-ST401, AusBiotech, Frenchs Forest, NSW, Australia), is passed around the coracoid and clavicle. In order to pass the tendon allograft, two loop sutures (FiberLink SutureTape 0.9 mm, with Loop, AR-7559, Arthrex, Naples, FL, USA) are passed from medial to lateral under the body of the coracoid process using a Dechamps suture passer (AC Wire Passer, AR-2252, Arthrex, Naples, FL, USA). The loop ends of the sutures are both placed laterally. Thereafter, two more loop sutures are passed behind the clavicle from inferior to superior. The loop ends of the sutures are both positioned inferiorly.

The tendon allograft is prepared using a whip stitch (SD wire loop AR-4068-05SD, Arthrex, Naples, FL, USA). The whip stitch is then connected to the tail of a silicon tissue dilator (Flexible obturator, AR-2275, Arthrex, Naples, FL, USA). This construct is then tied to one of the loop sutures previously passed around the coracoid process, and the tendon allograft is pulled into position following the path of the tissue dilator. Once passed under the coracoid process, the tissue dilator is tied to one of the loop sutures previously placed behind the clavicle, and the construct passes from inferior to superior so that the tendon allograft now runs posterior to the clavicle. The second loop suture from each pair is used to pass the tendon allograft a second time around the coracoid and clavicle. The tendon allograft is secured to itself using multiple simple sutures (2-0 FibreWire Suture, AR7220, Arthrex, Naples, FL, USA) (Figure 1C).

#### 2.1.4. ACJ Ligament Reconstruction

After the CC complex reconstruction, attention is turned to the AC ligament. The ACJ is reconstructed by drilling two 4.5 mm holes: the first through the lateral clavicle halfway between the Dog Bone and the AC joint and from anterior to posterior and the second in the acromion from anterior to posterior–superior (Figure 1D). The acromial tunnel must be positioned lateral enough to allow for the anchor for the internal brace construct to be placed between it and the AC joint (Figure 1D). A Hewson suture retriever (Art. No 71111579, Smith & Nephew, London, UK) is used to place a loop suture (FiberLink SutureTape 0.9 mm, with Loop, AR-7559, Arthrex, Naples, FL, USA) in the distal clavicle tunnel and another in the acromial tunnel.

#### 2.1.5. ACJ Internal Brace Repair

An AC repair is performed using an internal brace construct: a 3.5 mm Pushlock anchor (BioComposite Labral SwiveLock Anchor, 3.5 mm × 15.8 mm, AR-2325BCC, Arthrex, Naples, FL, USA) loaded with 2 mm Fibertape (AR-7237-7, Arthrex, Naples, FL, USA) in the acromion and a Swivelock 4.75 mm anchor (BioComposite SwiveLock C, 4.75 mm × 19.1 mm, AR-2324BCC, Arthrex, Naples, FL, USA) in the clavicle (Figure 1D).

Following the placement of the internal brace across the anterior aspect of the ACJ, a second tendon allograft is prepared peroneus longus (Peroneus Longus Tendon, AB-ST401 AusBiotech, Frenchs Forest, NSW, Australia). The tendon is split longitudinally to give a 4 mm diameter slip of tendon. One end of this tendon slip is placed into the end of a tendon passer (QuickPass Tendon Shuttle 3.5 mm, AR-8090L, Arthrex, Naples, FL, USA). The end of the tendon passer is placed into the loop of the loop suture and folded over on itself. The loop suture is then pulled through the tunnel with the tendon passer followed by the tendon allograft. A figure of eight construct is created with the tendon allograft over the top of the ACJ. The tendon allograft is secured with simple interrupted 2/0 FibreWire sutures (2-0 FiberWire Suture, AR-7220, Arthrex, Naples, FL, USA) (Figure 1E).

The temporarily positioned K-wire is removed. Lavage of the wound is performed with a 10% diluted Povodine-Iodine (Pfizer, Manhattan, NY, USA) saline solution. The wound is closed in layers, paying particular attention to the repair of the trapezius and deltoid over the superior aspect of the clavicle. A subcuticular 3/0 Monocryl (Art. No Y293H, Ethicon, Raritan, NJ, USA) is used to close the skin. A Dermabond Prineo skin closure system (Art. No ETH-CLR602US, Dermabond, Ethicon, Raritan, NJ, USA) is used to dress the incision, which is then covered by a waterproof dressing. A video summary of the described technique is available in the Appendix A.

#### 2.1.6. Post-Operative Management

Post-operatively, the shoulder is immobilized for 6 weeks in a shoulder support sling (DonJoy UltraSling III (Enovis, Lewisville, TX, USA)). Guided rehabilitation is commenced after 6 weeks with weaning of the sling, and a progressive resisted exercise program is instituted, allowing for passive stretching as required. Radiographs are performed at 6, 12, and 26 weeks post-operatively to confirm reduction is maintained. Elevation above 90° is not permitted until after review at 12 weeks post-op. After 12 weeks, full range of motion as tolerated is permitted. Full activity is not permitted until 6 months post-operatively.

### 2.2. Clinical Assessment

A retrospective case series on prospectively collected cases was conducted to support the technical note with functional and radiological outcomes. All consecutive adult patients who were treated surgically using this technique were included between May 2017 and May 2021 by an experienced shoulder surgeon (KC). All treated patients were included in this review. Patients suffering from a traumatic ACJ dislocation with at least a Rockwood IIIB without ipsilateral previous shoulder surgery were eligible for inclusion. Patients under the age of 18 years at the time of surgery and patients who were not able to provide informed consent were excluded. Informed consent was obtained from all participants under ethical approval from the Brisbane Private Hospital Human Research Ethics Committee (LREC19BPH9, 2022.14.374 and the Uniting Care Human Research Ethics Committee (2022.14.374).

#### 2.2.1. Radiographic Analysis

For radiographic analysis of vertical displacement, the pre-operative and 6-week post-operative coracoclavicular distances (CCD; in mm) were measured on unweighted anterior–posterior bilateral Zanca view radiography as the distance between the tip of the coracoid and the inferior cortex of the clavicle [20,21,22]. The side-to-side difference (in mm) in the CCD was obtained in relation to the non-injured contralateral side. Examples of pre- and post-operative X-rays have been included in Appendix A.

#### 2.2.2. Clinical Outcome Scores

Outcome measures were assessed pre-operatively and at 3 months, 6 months, and 12 months using scores for VAS pain, QuickDASH, Oxford Shoulder Score (OSS), Constant score (CMS), ASES and EQ5D, as well as a physical exam to examine forward flexion, lateral elevation, as well as internal and external rotation. Mean (SD) or medians (interquartile range (IQR)) are presented according to data normality and appropriate testing performed using SPSS for Windows (version 29, IBM Corp, Armonk, NY, USA). After confirmation of normality, paired *t*-tests were used to compare changes in scores and CCD at each time point as well as comparison with the contralateral normal shoulder. Significance tests were at the 5% level of significance, with no adjustments for multiple testing.

## 3. Results

Patient demographics are presented in Table 1. No additional pathology was identified during the arthroscopy, and no additional procedures were performed. Outcome scores were available for 11 patients (Table 2) at 12 months following surgery, with some missing scores prior to this time point—one patient did not complete any scores, pre- or post-operatively, so this patient is not included in the score analysis; however, there has been no further surgery or complications for this patient, and radiographic review was available. The radiological scores for all 12 patients at each time point are presented in Table 3.

The Constant Murley Score improved significantly from 27.6 (SD 16.9, range 8.0–56.5) to 61.5 (SD 15.0, range 42.0–92.0) at 12 months of follow-up (*p* = 0.006 paired *t*-test). There was one frozen shoulder post-operatively, with spontaneous recovery. There were no further complications observed with this technique. The CCD was significantly reduced from a mean of 18.7 mm (SD 3.4, range 13.0–24.0) before surgery down to 7.3 mm (SD 1.4, range 5–10.0; *p* = 0.007 paired *t*-test) on day 1 post-op, 10.0 mm (SD 3.4, range 6.0–16.0; *p* < 0.001 paired *t*-test) 12 weeks after surgery, and 10.5 mm (SD 2.0, range 8.0–14.0; *p* = 0.002 paired *t*-test) 12 months after surgery. There were no significant differences in CCD with the contralateral, uninjured shoulder at any time point post-operatively from day 1 onwards (*p* > 0.005 paired *t*-test, Table 3).

All PROMs improved significantly from pre-operatively by 3 months (*p* < 0.05 paired *t*-test) other than the EQ5D, which was not significantly different at 3 months (*p* = 0.128) but was significantly different from 6 months onward (*p* < 0.05 paired *t*-test). Range of motion took longer to improve significantly from pre-op levels, with both forward and lateral flexion improving significantly by 12 months (*p* < 0.05 paired *t*-test). Conversely, by 6 months post-op, forward and lateral flexion and external and internal rotation were no longer significantly different from the contralateral uninjured shoulder (*p* > 0.05 paired *t*-test).

## 4. Discussion

This novel technique, combining CC and AC repair with reconstruction and addressing horizontal as well as vertical stability in acute and chronic AC joint instability, is shown to be a viable treatment option based on our initial clinical and radiological results. By using this technique, it is believed that infero-superior and antero-posterior instability is addressed, with a potential lower risk of loss of reduction and residual shoulder complaints. The clinical and radiological results in this prospective cohort were promising, with satisfactory shoulder function and maintenance of the reduced coracoclavicular distance one year after surgery.

As previously mentioned, numerous stabilization techniques have been introduced, and multiple literature overviews have been provided comparing these techniques [2]. It has become clear that the use of K-wires and hook plates may be compromised by complications and inferior functional results [23]. Other techniques include free tendon grafting, suspensory devices, and Weaver–Dunn procedures, which showed comparable subjective results. Recently, arthroscopically assisted or full-arthroscopic CC and AC repair and reconstruction techniques have been developed and performed more frequently [24].

The majority of surgical techniques using tendon allografts or modifications of the Weaver–Dunn procedure stabilize the CC complex [11,25,26,27]. Nicholas et al. [25], in their study of nine patients with grade V AC joint dislocations, excised the distal 1 cm of the clavicle using an oscillating saw. They reconstructed the CC ligaments but did not reconstruct the AC ligaments. Our technique differs. We fixed both the AC and CC components and did not excise the distal clavicle end. Unlike them, our technique used a Dog Bone button fixation for addressing the vertical component of instability, which was augmented with a graft around the corocoid process. Our technique holds true for not just grade V but also grade IIIB, IV, V, and VI AC joint dislocations.

Scheibel et al. [26] described a surgical technique to address chronic AC joint dislocations. An autologous gracilis tendon was harvested from the ipsilateral side, which was then prepared. They addressed the CC complex by reconstructing it with the help of a TightRope fixation and a gracilis graft. They made two tunnels in the corocoid for the TightRope and the gracilis graft. We, on the other hand, fixed both the AC and CC components and did not use an autologous graft. By not using an autologous graft, the complications of donor-site morbidity were avoided. Also, we made only one single vertical tunnel in the corocoid, which we believe limits the possibility of stress risers, indirectly preventing late corocoid complications. Our technique can be used in cases of chronic and acute AC joint dislocations.

Boileau et al. [27], in their study, looked into 10 patients with grade III or VI dislocations who had previous surgeries on their AC joints and failed. They performed an arthroscopic Weaver–Dunn–Chuinard procedure with a double-button fixation. They had similar clinical outcomes compared to our study, but they noticed a few complications. Two patients had a delayed union of the excised transported acromion. One patient had a shift of the endobutton laterally off the corocoid, and one had a superficial infection. We did not transport or excise the acromion tip, and we did not see any complications other than a frozen shoulder that resolved spontaneously.

However, the ACJ capsule and ligaments have been shown to be highly important by Fukuda et al. [28] in preventing posterior subluxation of the distal end of the clavicle. Additionally, approximately 80% of the horizontal stability in this direction is provided by the intact posterosuperior AC ligaments, as demonstrated by Klimkiewicz et al. [15] in biomechanical studies of the ACJ complex. Furthermore, Rosso et al. reported the results of the 2020 ESA-ESSKA meeting in which the panel agreed on arthroscopically assisted anatomic reconstruction using a suspensory device (86.2%), with no need for a biological augmentation (82.8%) in acute injuries, whereas biological reconstruction of coracoclavicular and acromioclavicular ligaments with a tendon graft was suggested in chronic cases [29].

Martetschläger et al. [13] described an arthroscopically assisted technique to reconstruct the AC and CC ligaments using button suture tape (Dog Bone and FiberTape; Arthrex) in order to improve both horizontal and vertical stability [13]. Furthermore, in a cohort study of 26 patients with chronic AC joint injuries (Rockwood types 3–6), Tauber et al. [12] compared radiographic and clinical outcomes of adding AC ligament reconstruction to arthroscopic CC ligament reconstruction. The authors found that combined AC and CC ligament reconstruction restored horizontal stability, and patients undergoing combined reconstruction demonstrated improved clinical and radiological outcomes [12]. Nevertheless, their technique did not include a CC repair, and they stabilized the ACJ with a vertically drilled tunnel with the tendon graft sitting within the ACJ, functioning as a sort of ‘neo disc’. From an anatomical perspective, drilling these tunnels horizontally instead of vertically, as described in this novel technique, and creating a figure of eight configuration on top of the ACJ restores the posterior, superior, and anterior ligaments and capsule. Subsequently, the additional anterior-positioned internal brace supports the reconstruction as well. Another similar technique was described by Haber et al. [11], who performed a primary AC and CC reconstruction using two allografts, a TightRope repair of the CC and AC. However, they addressed the AC joint in a vertical fashion rather than horizontally, and unfortunately, no clinical results are reported to compare the outcome measures [11].

Concerning the onset of complaints and timing of surgery, various definitions have been proposed for defining an acute versus a chronic ACJ dislocation. The type of injury can be grossly divided into acute (0–3 weeks), subacute (3–6 weeks), and chronic (≥6 weeks) [30]. It remains a topic of debate whether to treat ACJ dislocations in the acute phase and if additional use of biologic reconstruction versus an isolated repair is indicated. It is strongly believed that, at least in chronic cases, the regenerative capacities of the capsuloligamentous structures of the ACJ are limited, which suggests the necessity of biologic tissue augmentation. In this study, acute and chronic cases were treated with the same technique.

Geraci et al. [31] treated 47 patients with acute grade III, IV, and V surgically with a ligament augmentation and reconstruction system (LARS) reconstruction. The overall complication rate was 4.2%. There were two cases, both male patients with a type III dislocation joint, wherein the LARS reconstruction failed approximately 20 days after the surgical treatment without having new reported trauma. Post-operatively, six of the 47 patients developed a keloid scar at the site of the surgical wound. We did not come across any such complications in our study. The LARS is an artificial ligament and thus reacts differently to the body’s normal biomechanical loading and stresses. The failure of the LARS could be attributed to quality issues rather than to the learning of a new technique. The long-term complications of a LARS reconstruction could occur due to stress shielding as it is an artificial structure. This will not be seen in our study as we did not use an artificial ligament.

It is important to note that, because this technique makes use of allografts, issues like taking a secondary incision for graft harvesting, graft site infection, possible weakness of internal rotation of the knee or eversion of the ankle after autograft removal, and other graft site morbidities do not arise. We encourage readers to continue to be cautious of graft hypersensitivity, infections due to the graft, inadequate quality of the graft for the procedure, etc., though we did not experience these complications in this series. Generally, allografts are available in all major hospitals and are typically subject to strict tissue bank policies, thus minimizing the frequency of graft-related infection. The cost of the allograft will vary according to the surgical setting, and this should be considered when offering this procedure to patients.

The surgical technique described in this study addresses the CC ligaments and the AC ligaments with biologic reinforcement. Based on the results showing excellent maintenance of the reduction and functional recovery, this technique could be considered a good alternative to the numerous existing techniques described previously.

This study did have limitations, which decreased the strength of our findings. Firstly, we had a small sample size of 12 patients, limiting the power of this study. Secondly, we had a heterogeneous group comprising both acute (6) and chronic (6) AC joint dislocations, limiting the applicability of our findings in both groups. Thirdly, we did not have any patients with grade VI AC joint dislocations. And lastly, we did not have access to a matched comparison group using an alternative technique.

## 5. Conclusions

This study describes a technique to treat acute and chronic Rockwood stage IIIB–V ACJ dislocations with promising short-term clinical and radiological results. The results suggest that the combined repair and reconstruction of the AC and CC ligaments is a safe procedure with low complication risk in experienced hands. Addressing the vertical and horizontal stability in ACJ dislocation is considered key to accomplishing optimal long-term results. Further follow-up is required to investigate this.

## Figures and Tables

**Figure 1 jcm-14-01730-f001:**
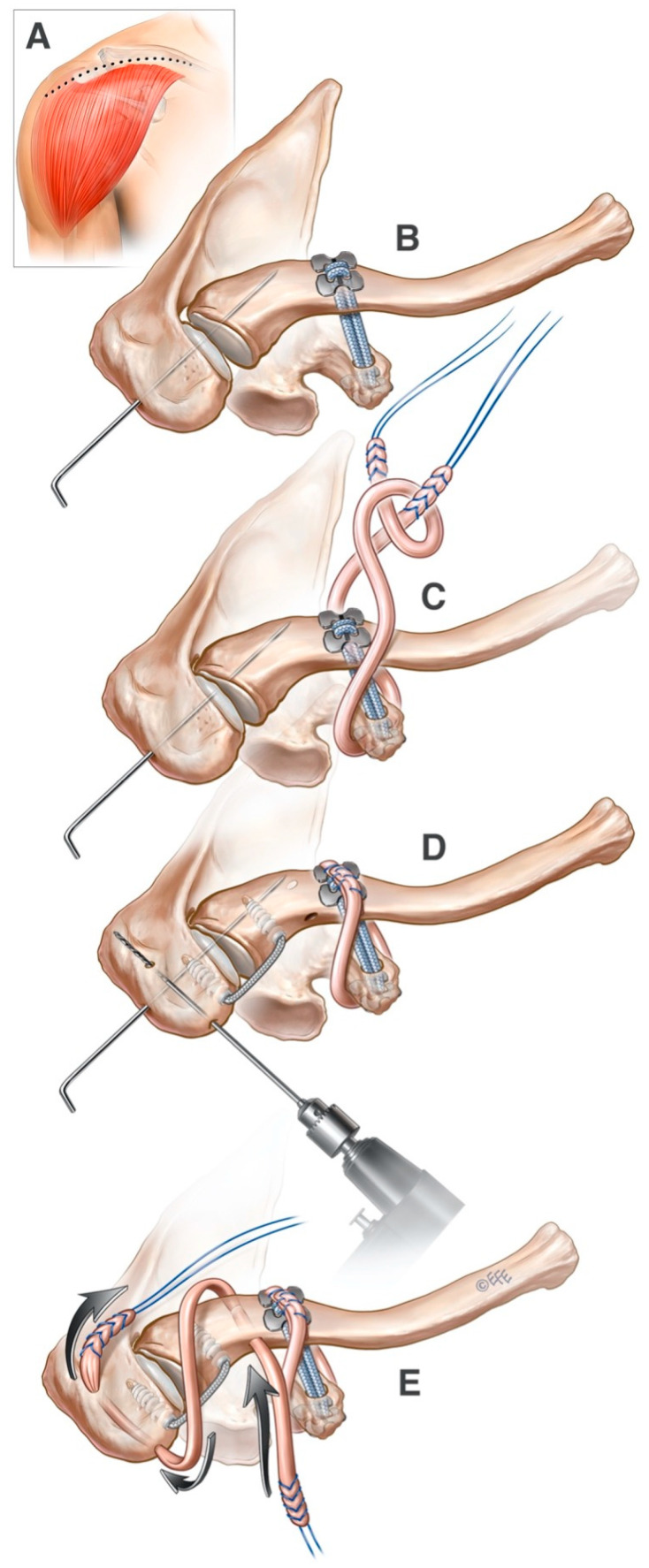
(**A**): superior clavicular approach with lateral extension; (**B**): AC reduction with temporary fixation and CC Dog Bone repair; (**C**): CC reconstruction with single-sided looped peroneal longus allograft; (**D**): AC repair using an internal brace; (**E**): AC reconstruction using a peroneal longus allograft and removal of K-wire to provide temporary reduction. If the tendon allograft used to reinforce the coracoclavicular ligament reconstruction is long enough, it may be passed twice around the coracoid and clavicle before securing it.

**Table 1 jcm-14-01730-t001:** Pre-operative characteristics (*n* = 12).

Variable	AC + CC Repair and Reconstruction
Age (mean (SD) years)	43.3 (SD 8.6)
Gender (male:female)	10:2
Acute:Chronic ^1^	6:6
Injury to surgery interval (median (IQR) days)	60.0 (IQR 127)
Concomitant shoulder injury	
rotator cuff tear	0
labral tear	3
SLAP tear	1
none	6
Concomitant intervention	
debridement small labral flap tear	1
labral repair/stabilization	2
biceps tenodesis	1
Rockwood type (I:II:IIIA:IIIB:IV:V:VI)	0:0:0:3:6:3:0

^1^ Acute: <6 weeks after trauma; chronic: >6 weeks after trauma.

**Table 2 jcm-14-01730-t002:** Clinical results.

Variable (Mean ± SD)	Pre-Operative *n* = 10	3 Months Post-Op *n* = 8	6 Months Post-Op *n* = 8	12 Months Post-Op *n* = 11	Contralateral Non-Injured Shoulder *n* = 7
VAS pain	66.1 ± 22.3	32.0 ± 23.7	13.5 ± 8.9	18.4 ± 24.5	N/A
Quick Dash	65.7 ± 22.0	34.4 ± 22.0	22.2 ± 17.6	12.0 ± 16.7	N/A
OSS	19.4 ± 11.3	33.8 ± 9.7	41.1 ± 6.0	42.6 ± 7.3	N/A
CMS	27.6 ± 16.9	36.2 ± 17.6	44.5 ± 16.5	61.5 ± 15.0	N/A
Forward flexion	113.3 ± 55.9	117.5 ± 34.5	168.8 ± 13.6	177.0 ± 9.5	180 ± 0.0
Lateral elevation	108.9 ± 6.03	108.1 ± 47.8	161.3 ± 19.6	175.1 ± 9.7	180 ± 0.0
GH ER1	44.8 ± 26.4	40.6 ± 20.4	48.1 ± 20.3	61.1 ± 15.3	77.1 ± 15.0
GH ER2	50.8 ± 43.8	45.0 ± 25.1	68.8 ± 16.4	84.5 ± 8.3	85.4 ± 13.4
GH IR1 ^1^	5.1 (range 0–10)	4.8 (range 0–10)	7.0 (range 4–10)	8.2 (range 6–10)	8.9 (range 8–10)
GH IR2	40.0 ± 29.6	44.4 ± 30.9	50.6 ± 28.6	65.5 ± 21.4	68.3 ± 35.1
EQ-5D health index	0.50 ± 0.30	0.75 ± 0.24	0.81 ± 0.19	0.87 ± 0.20	N/A
ASES total	31.6 ± 24.1	57.2 ± 23.2	77.8 ± 13.2	89.2 ± 12.7	62.8 ± 15.1

^1^ IR 1: 0 = Dorsum of hand to lateral thigh; 2 = buttock; 4 = lumbosacral junction; 6 = waist L3; 8 = T12; 10 = interscapular area (T7) (range reported); OSS = Oxford Shoulder Score, CMS = Constant Murley Score, GH = Glenohumeral, ASES = American Shoulder Elbow Surgeons Score, N/A = Not Applicable.

**Table 3 jcm-14-01730-t003:** Radiological results (compared using paired *t*-test).

Variable (Mean (SD))	Pre-Op *n* = 11	1 Day Post-Op *n* = 10	6 Weeks Post-Op *n* = 12	3 Months Post-Op *n* = 12	6 Months Post-Op *n* = 12	12 Months Post-Op *n* = 8	Non-Injured Shoulder *n* = 9 (1 Injured, 2 Missing)
X-ray							
CCD	18.7 (SD 3.4)	7.3 (SD 1.4)	9.4 (SD 3.0)	10.0 (SD 3.4)	11.1 (SD 3.0)	10.5 (SD 2.0)	10.0 (SD 2.4)
CCD difference with uninjured shoulder	9.1 (SD 3.6) *p* < 0.001	1.7 (SD 1.1) *p* = 0.007	0.0 (SD 3.5) *p* = 1.0	0.8 (SD 2.7) *p* = 0.410	1.9 (SD 2.5) *p* = 0.055	0.6 (SD 1.8) *p* = 0.436	

CCD = Coracoclavicular distance.

## Data Availability

Research data are available upon request. Please direct inquiries to the corresponding author.

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
