# Peer review of "Combined Repair and Reconstruction of Coracoclavicular and Acromioclavicular Ligaments for Acute and Chronic AC Joint Dislocations: A Technical Note and Prospective Case Series"

_jcm, 2025, doi:10.3390/jcm14051730_

Round 1

Reviewer 1 Report

Comments and Suggestions for Authors

This manuscript presents a novel surgical technique for managing acute and chronic AC joint dislocations, combining CC ligament repair and AC ligament reconstruction. The study includes a prospective case series of 12 patients. The technique addresses both vertical and horizontal stability of the AC joint, suggesting its efficacy and safety.
overall this study provide a new technique of reconstruction among many previous techniques
However, the big limitation that the study lack of comparison with other techniques and sample size is too low. Moreover, the heterogenicity between group (Acute + chronic) 

Some comments to be addressed 

i suggest to change the title to "Combined Repair and Reconstruction of Coracoclavicular and Acromioclavicular Ligaments for Acute and Chronic AC Joint Dislocations: A Technical Note and Prospective Case Series". >> if you want that this article is a technical note form , so it's not acceptable as original article

increase the discussion comparing other techniques 

Discuss the pros and cons of this technique more in details >> availabilty of allograft , cost , infection, morbidity of surgical incision in comparison with arthroscopic techniques

If authors have a matched group with other technique , it improve the quality of the manuscript

statistical analysis should more be detailed , which test was used to measure p -value ? 

how many patients had complications such as delay wound healing, infection , keloid scar etc ? 
also surgical time , risk of

add limitation of the study , specially that samples are heterogenous (acute+chronic)  and low sample size, absence of comparative group 

Author Response

Reviewer #1

This manuscript presents a novel surgical technique for managing acute and chronic AC joint dislocations, combining CC ligament repair and AC ligament reconstruction. The study includes a prospective case series of 12 patients. The technique addresses both vertical and horizontal stability of the AC joint, suggesting its efficacy and safety.

overall this study provide a new technique of reconstruction among many previous techniques However, the big limitation that the study lack of comparison with other techniques and sample size is too low. Moreover, the heterogenicity between group (Acute + chronic)

Some comments to be addressed

I suggest to change the title to "Combined Repair and Reconstruction of Coracoclavicular and Acromioclavicular Ligaments for Acute and Chronic AC Joint Dislocations: A Technical Note and Prospective Case Series". >> if you want that this article is a technical note form , so it's not acceptable as original article

Thank you for your input. The authors agree and have changed the manuscript title as suggested.

Title changed to: Combined Repair and Reconstruction of Coracoclavicular and Acromioclavicular Ligaments for Acute and Chronic AC Joint Dislocations: A Technical Note and Prospective Case Series

Increase the discussion comparing other techniques

The authors agree and have expanded the discussion to include comparisons to other techniques, highlighting points of difference with respective studies.

Lines 279 – 316 – Added comparisons to previously published techniques and discussed associated points including; repair without allograft re-enforcement, use of autografts, repair of CC complex only, etc.

Discuss the pros and cons of this technique more in details >> availability of allograft, cost, infection, morbidity of surgical incision in comparison with arthroscopic techniques

The authors note that both availability and cost of the allograft will vary according to the area of practice. In this instance we have elected to focus on the clinical outcomes and technical description of the technique. As such, we have not discussed factors surrounding the availability, or lack thereof, of allografts and their associated costs in detail, as we believe this would be better highlighted in a separate publication.

We did not have any complications (infection, etc.) other than a frozen shoulder that resolved spontaneously, as reported in the manuscript (Lines 236 – 237). Thus, we have not elaborated on the complications.

The authors would suggest that even in an arthroscopic procedure, one needs to make an incision over the clavicle region. Our incision was not bigger than required to perform adequate exposure for surgery. Thus we did not experience any morbidity of the surgical incision.

If authors have a matched group with other technique , it improve the quality of the manuscript

The authors agree that this comparison would lead to a stronger study. Unfortunately, due to limitations associated with the single site nature of this study, the authors do not have access to a matched comparison group. As such, we have not been able to conduct this analysis.

statistical analysis should more be detailed , which test was used to measure p -value ?

The authors agree and have expanded the methods section to include details relating to the statistical analysis. We have also the test used when a p value is reported.

Lines 222 – 225 - After confirmation of normality, Paired t-tests were used to compare changes in scores and CCD at each time point, as well as comparison with the contralateral normal shoulder. Significance tests were at the 5% level of significance, with no adjustments for multiple testing.

Lines 243 – 250 – Added paired t-test identifier for reported p values.

how many patients had complications such as delay wound healing, infection , keloid scar etc ? also surgical time , risk of

The authors did not witness any complications other than a frozen shoulder that resolved spontaneously, which has been reported in the manuscript (Lines 236 – 237).

add limitation of the study , specially that samples are heterogenous (acute+chronic)  and low sample size, absence of comparative group

The authors agree and have expanded the discussion to incorporate the limitations section.

Lines 359 – 364 - The study did have limitations, which decreased the strength of our findings. Firstly, we had a small sample size of 12 patients, limiting the power of the study. Secondly, we had a heterogeneous group comprising of both acute (6) and chronic (6) AC joint dislocations, limiting the applicability of our findings in both groups. Thirdly, we did not have any patients with grade VI AC joint dislocations. And lastly, we did not have access to a matched comparison group using an alternative technique.

Reviewer 2 Report

Comments and Suggestions for Authors

This study was very interesting, and I gained a lot of insights from reading it. I have a few comments for improvement that I hope will be helpful.

1.     The current keywords are broad and general. Adding more specific and precise terms would better reflect the focus of the study.

2.     Line 230 and 247: Please confirm if “p>0.005” is a typo and should be “p>0.05”. If “p>0.005” is correct, clarify why this significance level was used. Also, include the defined significance level (e.g., p=0.05) in the Methods section.

3.     It would be helpful to include surgery time information. If possible, add this to Table 1.

4.     This study introduces a new surgical technique for acute and chronic ACJ dislocations. However, the novelty of the technique is not clear. Please explain how this technique is better than existing methods, such as reduced complications, faster recovery, or lower technical difficulty.

5.     The study describes a technique using an allograft, but allografts are not available in some regions, such as Japan. Please discuss the differences in outcomes and techniques when using artificial ligaments or strong sutures instead of allografts. It would also help to explain the advantages and disadvantages of allografts compared to artificial materials.

6.     There is no mention and limitations or future studies in the paper. Adding these sections would strengthen the discussion.

7.     In the Conclusion, compare this technique with existing methods and highlight its clinical benefits. This will clarify the position and importance of your research.

Author Response

Reviewer #2

This study was very interesting, and I gained a lot of insights from reading it. I have a few comments for improvement that I hope will be helpful.

The current keywords are broad and general. Adding more specific and precise terms would better reflect the focus of the study.

Thank you for pointing this out. The authors agree and have edited our keywords to the following - AC joint reconstruction; AC joint repair; AC and CC ligament complex; Acute and Chronic AC joint dislocation; Technical note

Line 230 and 247: Please confirm if "p>0.005" is a typo and should be "p>0.05". If "p>0.005" is correct, clarify why this significance level was used. Also, include the defined significance level (e.g., p=0.05) in the Methods section.

Thank you for pointing this out. P>0.005 was a typo and we have corrected it in the manuscript to p>0.05. We have also clarified the significance level in the methods section.

Lines 244 – 251 – Corrected p<0.005 to p<0.05 in line with established statistical significance threshold.

Lines 224-225 - Significance tests were at the 5% level of significance, with no adjustments for multiple testing.

It would be helpful to include surgery time information. If possible, add this to Table 1.

Unfortunately the authors do not have access to data on the exact duration of the surgical procedures included in this study. To the best of our knowledge, procedures took between one and one and a half hours, though this estimate is entirely anecdotal.

This study introduces a new surgical technique for acute and chronic ACJ dislocations. However, the novelty of the technique is not clear. Please explain how this technique is better than existing methods, such as reduced complications, faster recovery, or lower technical difficulty.

Thank you for your suggestion. Unfortunately, due to a lack of a matched comparative group, we are unable to draw reliable conclusions regarding the superiority or inferiority of the described technique. We instead present our findings in the hope that readers will compare our results to those of their own practice, or previously published findings. We have expanded the discussion section to include additional comparisons to alternative techniques where possible.

We believe that this technique, combining CC and AC repair with reconstruction and addressing horizontal as well as vertical stability in acute and chronic AC joint instability, has shown to be a viable treatment option based on our initial clinical and radiological results.

The study describes a technique using an allograft, but allografts are not available in some regions, such as Japan. Please discuss the differences in outcomes and techniques when using artificial ligaments or strong sutures instead of allografts. It would also help to explain the advantages and disadvantages of allografts compared to artificial materials.

The authors agree with your feedback and have expanded the discussion to include comparison to other such techniques, highlighting points of difference in the manuscript.

Lines 279 – 316 – Added comparisons to previously published techniques and discussed associated points including; repair without allograft re-enforcement, use of autografts, repair of CC complex only, etc.

Lines 345 – 355 – Added to the discussion with comparison to a LARS reconstruction using an artificial ligament.

There is no mention and limitations or future studies in the paper. Adding these sections would strengthen the discussion.

The authors agree and have expanded the discussion to incorporate study limitations.

Lines 360 – 367 - The study did have limitations, which decreased the strength of our findings. Firstly, we had a small sample size of 12 patients, limiting the power of the study. Secondly, we had a heterogeneous group comprising of both acute (6) and chronic (6) AC joint dislocations, limiting the applicability of our findings in both groups. Thirdly, we did not have any patients with grade VI AC joint dislocations. And lastly, we did not have access to a matched comparison group using an alternative technique.

In the Conclusion, compare this technique with existing methods and highlight its clinical benefits. This will clarify the position and importance of your research.

Thank you for your feedback. We believe we have accomplished this, as outlined in our above responses.

Round 2

Reviewer 1 Report

Comments and Suggestions for Authors

Dear authors, thank you for your corrections

i still have some points that could ameliorate the manuscript 

1- similarity index is high 29% , need to be less than 20 % 

2- provide pre-op and post op x-ray if available

3- provide per-op images if available

4- regarding my last comment "Discuss the pros and cons of this technique more in details >> availability of allograft, cost, infection"

i thank you for your response, however if you don't mind, i believe this details still important even if it's written in 1 small paragraph ( and even if there was no infection) 

i have no other comment to add, 

Author Response

Reviewer 1 – Round 2

Dear authors, thank you for your corrections

I still have some points that could ameliorate the manuscript 

1- similarity index is high 29%, need to be less than 20 % 

Thank you for this feedback. Reviewing the iThenticate report, we note that the majority of the flagged sections contain language that is commonly seen in a wide variety of publications. The report also highlights portions of the conflict-of-interest declaration, funding, IRB, and informed consent statements, which are boilerplate templates included at the request of JCM.

While the authors are open to re-wording the highlighted sections, we feel that we would be doing so to meet an artificial target for the similarity index, while compromising the readability and understandability of the manuscript.

While we have not made changes based on this recommendation, if there are ongoing concerns regarding plagiarism of specific portions of the manuscript, we are of course more than willing to discuss and address them.

2- provide pre-op and post op x-ray if available

A pre and post op x-ray for a participant has been added to the supplementary materials. Additional x-rays are available for all participants upon request.

Lines 213 – 214 - Examples of pre and post-operative x-rays have been included in supplementary file 2.

3- provide per-op images if available

Unfortunately, per-op images are not available for each procedure. We encourage readers to review the technique video included in the supplementary materials, which may serve as an alternative.

4- regarding my last comment "Discuss the pros and cons of this technique more in details >> availability of allograft, cost, infection"

i thank you for your response, however if you don't mind, i believe this details still important even if it's written in 1 small paragraph (and even if there was no infection) 

Thank you for your feedback. We have added a section to the discussion to reflect this.

Lines 357 – 366 - It is important to note that because this technique makes use of allografts, issues like taking a secondary incision for graft harvesting, graft site infection, possible weakness of internal rotation of the knee or eversion of the ankle after autograft removal, and other graft site morbidities, do not arise. We encourage readers to continue to cautious of graft hypersensitivity, infections due to the graft, inadequate quality of graft for the procedure, etc., though we did not experience these complications in this series. Generally, allografts are available in all major hospitals and are typically subject to strict tissue bank policies, thus minimizing the frequency of graft related infection. The cost of the allograft will vary according to the surgical setting, and this should be considered when offering this procedure to patients.

i have no other comment to add, 

Thank you for your thorough review and feedback on this manuscript.